# Enhancing Anti-Tumorigenic Efficacy of Eugenol in Human Colon Cancer Cells Using Enzyme-Responsive Nanoparticles

**DOI:** 10.3390/cancers15041145

**Published:** 2023-02-10

**Authors:** Nisitha Wijewantha, Sanam Sane, Morgan Eikanger, Ryan M. Antony, Rashaun A. Potts, Lydia Lang, Khosrow Rezvani, Grigoriy Sereda

**Affiliations:** 1Department of Chemistry, The University of South Dakota, 414 E. Clark Street, Vermillion, SD 57069, USA; 2Division of Basic Biomedical Sciences, Sanford School of Medicine, The University of South Dakota, 414 E. Clark Street, Lee Medical Building, Vermillion, SD 57069, USA

**Keywords:** drug-delivery system, nanoparticles, eugenol, colon cancer cells, apoptosis

## Abstract

**Simple Summary:**

The migratory and invasive pathways that evolve in the early stage of colorectal cancer (CRC) are crucial in developing the metastatic cascade and drug resistance. Natural plant-based compounds and their active secondary metabolites selectively target several oncogenic and oncosuppressive signaling pathways. The anticancer mechanisms medicated by plant-based compounds can circumvent the associated side effects observed with chemotherapeutic agents. This study has developed a new potential anticancer therapeutic: casein-coated nanoparticles (NPs) encapsulating eugenol (EUG), a potent anti-metastatic molecule. The active form of matrix metalloproteinases (MMPs) predominantly releases at the leading edges of migrating tumor cells and can locally digest the “gatekeeping” casein coat of the EUG-carrying NPs and expose CRC cells to a high local concentration of EUG. Therefore, the targeted delivery of EUG by a “smart” nanoparticle can significantly improve its therapeutic index and minimize side effects on normal cells.

**Abstract:**

This study is focused on the selective delivery and release of the plant-based anticancer compound eugenol (EUG) in colorectal cancer cells (CRC). EUG is an apoptotic and anti-growth compound in diverse malignant tumors, including CRC. However, EUG’s rapid metabolization, excretion, and side effects on normal cells at higher dosages are major limitations of its therapeutic potential. To address this problem, we developed a “smart” enzyme-responsive nanoparticle (eNP) loaded with EUG that exposes tumors to a high level of the drug while keeping its concentration low among healthy cells. We demonstrated that EUG induces apoptosis in CRC cells irrespective of their grades in a dose- and time-dependent manner. EUG significantly decreases cancer cell migration, invasion, and the population of colon cancer stem cells, which are key players in tumor metastasis and drug resistance. The “smart” eNPs–EUG show a high affinity to cancer cells with rapid internalization with no affinity toward normal colon epithelial cells. NPs–EUG enhanced the therapeutic efficacy of EUG measured by a cell viability assay and showed no toxicity effect on normal cells. The development of eNPs–EUG is a promising strategy for innovative anti-metastatic therapeutics.

## 1. Introduction

Natural plant extracts such as camptothecin [1] and vinblastine [2] are examples of successful anticancer agents entering the clinic [3]. Naturally derived anticancer compounds and their metabolites can target different signaling pathways and cellular mechanisms while generating fewer side effects than current genotoxic drugs such as 5-Fluorouracil (5-FU) [3,4]. However, plant-based molecules also have adverse effects which limit their therapeutic benefits [5]. Nanotechnology has become an effective and selective local drug delivery tool for cancer treatments in the last decade [6]. The release of natural compounds from their enzyme-responsive nanoparticle carriers (eNPs) can be predictively and selectively triggered by specific enzymes expressed in tumor tissues, resulting in the safe and effective targeted delivery of anti-tumor drugs. Therefore, the eNPs can reduce systemic toxicity while enhancing the drug’s therapeutic effect due to the precisely engineered concentration pattern of the drug in the body [7].

Eugenol, 2-methoxy-4-(2-propenyl)phenol, is a naturally occurring phenolic compound that is one of the main components in clove oil and honey [8]. Eugenol is traditionally used in Asian medicine due to its biological and medicinal properties, where it has been employed to treat dental and digestive disorders [9] alongside its use as antimicrobial [10], antioxidant [11], anti-inflammatory, antiseptic, and analgesic agents [12]. The use of EUG in humans has been approved by the Food and Agriculture Organization (FAO) and World Health Organization (WHO) [13]. Furthermore, recent reports revealed that EUG had demonstrated anticancer activity against several cancers, including liver, colon, and breast, where EUG prevents cancer progression via modulating pathways implicated in cell growth, apoptosis, and angiogenesis [9,10,12,14]. However, whether EUG can suppress key elements contributing to tumor metastasis and drug resistance in CRC remains unknown.

EUG shows a rapid metabolization and excretion in the urine within 24 h [15]. In normal cells, oxidative stress and DNA damage induced by EUG, particularly at high concentrations, have been two major limiting factors for the EUG’s therapeutic benefits in patients with cancer [16,17,18]. The short bioavailability and adverse effects have halted EUG from developing into a potent plant-based anticancer therapy in patients with cancers, including CRC. Pharmacokinetics of eugenol as a direct activator of dopamine have been studied in rat brains [19].

The limitations of eugenol can be alleviated by its targeted delivery by nanomaterials. While delivery of eugenol by solid lipid nanoparticles [20], calcium citrate nanoparticles [21], tannic acid-assisted cross-linked nanoparticles [22], and calcium carbonate- and hydroxyapatite-based particles [23] is known, neither of those formulations releases eugenol on-demand when triggered by an enzyme.

We developed a mesoporous silica nanoparticle-based system designed to deliver an anticancer drug molecule to target CRC cells selectively. Among many other drug-delivery nanocarriers, mesoporous silica nanoparticles (MSNs) have gained significant recognition in the last two decades [24,25,26,27] due to the MSN’s well-ordered internal mesoporous structure (typically ca. 2–6 nm) with large pore volume (0.6–1 cm^3^/g), high surface area (700–1000 m^2^/g), and tunable size (50–200 nm) [28,29,30,31,32]. High drug loading capacity, presence of gate-like scaffolds, easy surface modification, shape, and robustness are ideal for using MSNs in site-specific drug delivery systems [25,33,34]. Moreover, zero premature release is a remarkable property of end-capped MSN drug delivery systems [35]. The controlled drug release is powered by the ability of nanoparticles to hold drug molecules inside their pores capped by covalently bound “gatekeeping molecules.” These “gatekeeping molecules” are detached by stimuli at the specific target site, triggering the drug release [31]. Among other stimuli-responsive drug release systems, enzyme-responsive, controlled-release delivery systems show promise due to their biocompatibility and quick response to biological stimuli [36,37]. Cancer cells, including metastatic CRC tumor cell lines, release a specific set of matrix metalloproteinases (MMPs) necessary for their cell invasion [38]. Active MMPs are predominantly released at the leading edges of migrating tumor cells [39] and are absent in normal tissues [40]. Digestion of the “gatekeeping element” of nanoparticles by MMPs selectively released by cancer cells turns MSNs into a safe and effective drug delivery system. Finally, MSNs can increase the bioavailability of their cargo by surface modification, increasing their hydrophilicity [41]. We selected casein as the “gatekeeping element”, which seals eugenol inside nanoparticles and releases it when digested by the MMP enzymes overproduced by many types of cancerous cells.

This study reveals that EUG interferes with cancer cell migration and invasion while simultaneously decreasing the population of colon cancer stem cells (CSCs). We designed an enzyme-responsive drug delivery system, “EUG/CAS–MSNs–COOH,” that can carry anticancer agents (EUG) and selectively deliver them directly to the tumor site. Casein is covalently attached to the MSNs by coupling its modified surface’s carboxy groups with casein’s amino groups (Figure 1). The EUG loading capacity of the nanoassembly was investigated both in vitro and at the cellular level. EUG/CAS–MSNs–COOH was introduced as a promising drug delivery system for targeted primary and metastatic CRC.

## 2. Materials and Methods

### 2.1. Cell Culture

The colon cancer cells (HCT-116, SW480, SW620, HT29) and FHC (CRL-1831) human normal colon epithelial cell lines were purchased from American Type Culture Collection (Manassas, VA, USA) and were grown in the recommended medium. All examined cell lines were in passages limited to 10 and, per the manufacturer’s protocol, were routinely checked for mycoplasma contamination using the PCR mycoplasma detection kit (Applied Biological Materials Inc. (abm), Richmond, BC, Canada).

### 2.2. Western Blot, Immunofluorescent and Crystal Violet Staining Method Assays

Cell lysates were prepared using a digitonin lysis buffer (50 mM Tris/HCl, pH 7.5, 150 mM NaCl, 1% Digitonin (Sigma-Aldrich, St. Louis, MO, USA) plus 1× mammalian complete protease inhibitor (Research Products International Corp., Mount Prospect, IL, USA). Following homogenization (two hours of gentle rocking at 4 °C), cell lysates were centrifugated at 13,000 RPM for 10 min at 4 °C, and the supernatant was transferred to a fresh tube in preparation for Western blot (WB). Cell lysates used in WB were normalized for equal loading with NanoDrop using direct absorbance at 280 nm (ThermoFisher Scientific, Waltham, MA, USA). The samples were loaded onto SDS-PAGE 4–20% gradient gel, and the protein transfer was performed using an iBlot 2 system for probing with the corresponding antibodies: anti-MMP7 (Santa Cruz MMP-7 Antibody, catalog number: sc-515703, dilution 1:500) and β-tubulin (Proteintech, catalog number: 66240-1-Ig, dilution 1:2000). Immunofluorescent assays were conducted as previously described [42]. The cytotoxicity test protocol using a crystal violet staining method has been described previously [43] using a commercial kit provided by Abcam (Crystal violet Assay Kit or Cell viability, catalog number#ab232855). We used crystal violet staining to compare the entire stained cells per plate between groups. The measurement of absorption of the entire plates post =-staining with the crystal violet provided a more highly accurate reading than the manual counting of the cells under a microscope. Absorption of cells per well were measured with an ELx808 plate reader (BioTek, Santa Clara, CA, USA).

### 2.3. Flow Cytometry Analysis

A BD Accuri™ C6 plus cell cytometer (BD Biosciences, Franklin Lakes, NJ, USA) with 488 nm and 640 nm excitation lasers was used for all flow cytometry analysis according to the manufacturer’s standard protocols (BD Biosciences, San Jose, CA, USA). Colon cancer cell lines were analyzed using flow cytometry to determine the apoptotic markers (propidium iodide (PI), Annexin, and caspase 3) and the population of CD44^+^, CD133^+^, and Lgr5^+^ following respective antibody staining protocols provided by the manufacturers (BD Biosciences and Novus Biological). See Appendix A for a detailed list of fluorescent antibodies used in this experiment. An intracellular staining flow cytometry kit (catalog number: BP2-29450) and a cell surface staining flow cytometry kit (catalog number: nbp2-26247) were purchased from Novus Biologicals, USA. Data were analyzed using FlowJo v9 (FlowJo, BD, USD).

### 2.4. xCELLigence Real-Time Cell Analysis

xCELLigence Real-Time Cell Analyzer (RTCA, Agilent, Santa Clara, CA, USA) system technology measured cell migration, invasion, and adhesion assays in real-time, as previously described [44]. GraphPad Prism (version 9) was used to analyze the rate of cell proliferation (slope) during a critical time window for the HCT-116 cell line in the presence and absence of 10% FBS in McCoy’s 5a Medium. Cells treated with Dimethylsulfoxide (DMSO) or EUG were immediately loaded (2 × 10^4^ cells) into gold microelectrode plates. E-plates were used for the cell adherent assay, and CIM-plates ± Matrigel were used for migration and invasion assays. The xCELLigence system was used to measure the impedance value of each well every 15 min for 120 h. The outcome of measurements is a CI (cell index) value for cell adhesion, migration, and invasion. Plots were used to illustrate the CI value for overtime points. As previously described [45], the critical time point for cell adhesion, migration, and invasion was approximately 50 to 70 h after starting the experiments.

### 2.5. Reagents for Nanoparticle Assembly

Cetyltrimethylammonium bromide (CTAB), tetraethyl orthosilicate (TEOS, 99.98%), (3-aminopropyl) triethoxysilane (APTES), and eugenol (4-allyl-2-methoxyphenol) were purchased from Sigma-Aldrich. Casein, trypsin solution, absolute ethanol (EtOH), succinic anhydride, 1-ethyl-3-(3-dimethyl-aminopropyl) carbodiimide hydrochloride (EDC), *N*-hydroxysulfosuccinimide sodium salt (sulfo-NHS), *N*,*N*-dimethylformamide (DMF), 2-(*N*-morpholino)ethanesulfonic acid (MES) buffer, and glacial acetic acid were purchased from ThermoFisher Scientific, USA. Chemical reagents were of analytical grade and were used as received.

### 2.6. Instrumentation

A scanning electron microscope [SEM, model SIGMA FE-SEM (Zeiss)] and transmission electron microscope (TEM) were used to characterize the morphology and size of the prepared particles. The powder X-ray diffractometer (XRD) (model Rigaku, ultima lV) was used to analyze the phase distribution and structure of the prepared powder materials. Fourier transform infrared (FT-IR) spectroscopy (Bruker IFS Equinox 55 Spectrometer) was carried out to probe a functionalized particle surface. The particles’ zeta potential values and hydrodynamic diameters were obtained using the Malvern Zetasizer ZS nano series. The delivered drug concentration was monitored with UV spectrometry using a UV-500 UNICAM spectrometer.

### 2.7. Nanoparticle Synthesis and Surface Modification

Mesoporous silica nanoparticles (MSNs) were synthesized and surface-modified according to the procedures that we have previously reported [46]. Briefly, 2.00 g of CTAB was stirred in nanopure water, and NaOH (7 mL, 2.00 M) was added to the solution while adjusting the solution temperature to 80 °C. TEOS (11.477 mL) was added dropwise (0.5 mL/min) to the surfactant solution and stirred continuously for another 2 h to produce a white precipitate. The resulting nanoparticles were collected by centrifugation and washed three times with ethanol followed by three times with nanopure water before being dried at 60 °C overnight in an oven to produce as-synthesized MSN particles. The oven-dried product was calcined at 550 °C in a furnace in air for 2 h to obtain MSN particles (1.95 g). As previously described, the carboxylation of MSNs was performed in two steps [46]. First, MSNs were aminated by reaction with (3-aminopropyl) triethoxysilane (APTES). Then, 1.00 g of MSNs was dispersed in 30 mL of absolute ethanol, followed by adding 1 mL of APTES and 0.6 mL of glacial acetic acid to the mixture. After slowly stirring the mixture for 24 h at room temperature (RT), the resulting product was separated by centrifugation, washed with absolute ethanol three times, and dried at 80 °C in a vacuum oven for 24 h to obtain aminated MSNs (MSN-NH_2_). In the second step, the amino groups were converted into carboxylic acid groups by treating aminated MSNs with succinic anhydride. MSN-NH_2_ particles (200 mg) were mixed with 25 mL of a 1 mg/mL solution of succinic anhydride in DMF. The mixture was gently stirred for 24 h at RT. The resulting product was isolated by centrifugation and washed with DMF three times. The separated product was dried at 50 °C in a vacuum oven (700 mmHg) for 24 h to obtain MSN–COOH particles (195 g). 

### 2.8. Eugenol (EUG) Loading and Synthesis of EUG/MSN and EUG/MSN–COOH

Eugenol was loaded to both MSN and MSN–COOH particles. The MSN or MSN–COOH particles (5 mg) were soaked in a 1.5 mL eugenol solution (PBS buffer, pH 7.4) at given concentrations and dispersed on a rotary mixer for 24 h at RT. The obtained solution was centrifuged (11,000 RPM, 8 min) to remove unloaded EUG. The EUG-loaded particles were washed three times with 1.5 mL PBS buffer (pH 7.4, 0.1 M) to remove the surface adsorbed EUG, resulting in the production of EUG/MSNs and EUG/MSNs–COOH from MSNs and MSNs–COOH, respectively. To determine drug encapsulation efficiency (DEE) and drug loading efficiency (DLE) of EUG for particles, all washing supernatants were collected, and the amount of free EUG was determined spectrophotometrically at λ_max_ 280 nm with the aid of a calibration curve of known concentrations of EUG solutions. The DEE and DLE of eugenol for particles were calculated using Equations (1) and (2).
(1)$$\mathrm{DEE}=\frac{\mathrm{fed}\;\mathrm{drug}\;\left(µg\right)-\;\mathrm{drug}\;\mathrm{in}\;\mathrm{the}\;\mathrm{supernatant}\;(µg)}{\mathrm{fed}\;\mathrm{drug}\;(µg)}\;\times \;100$$
(2)$$\mathrm{DLE}=\frac{\mathrm{fed}\;\mathrm{drug}\;\left(µg\right)-\;\mathrm{drug}\;\mathrm{in}\;\mathrm{the}\;\mathrm{supernatant}\;(µg)}{\mathrm{mass}\;\mathrm{of}\;\mathrm{the}\;\mathrm{dried}\;\mathrm{particles}\;(µg)}\times 100$$

### 2.9. Conjugation of Casein Protein to EUG/MSNs–COOH

To covalently conjugate casein with the nanoassembly, EUG/MSNs–COOH (5 mg) were dispersed in 1 mL of MES buffer (0.1 M, pH 6.0). Next, 24 μL of 250 mM EDC in nanopure water and 240 μL of 250 mM sulfo-NHS (in 0.1 M MES buffer, pH 6.0) were quickly added to the EUG/MSNs–COOH suspension. The suspension was incubated by mixing on a rotary wheel for 30 min at room temperature (RT). The resulting particles were centrifuged (8000 RPM, 8 min) and gently washed with 1 mL MES buffer (pH 6.0) to remove excess EDC and sulfo-NHS. The collected particles were redispersed in 400 μL of PBS buffer (0.1 M, pH 7.4), followed by the addition of casein (1 mL of 0.1 M PBS buffer, pH 7.4) solution at given concentrations (0.1%, 0.2%, 0.3%, 0.4%, 0.5%, 0.6%, 0.7%, and 0.8%, *w*/*v*). The suspension was mixed for 5 h on a rotary mixer at RT to attach amino groups of casein to the carboxylic groups of EUG/MSNs–COOH. Finally, casein-capped drug-loaded particles (EUG/CAS-MSNs–COOH) were centrifuged and rinsed with PBS buffer (pH 7.4) twice to remove the excess casein. The combined supernatants were analyzed using a BCA assay to determine the extent of conjugation with casein.

### 2.10. In Vitro Eugenol (EUG) Release Measurements

In vitro stimuli-responsive release of EUG from the casein-capped nanoassemblies without (control series) and in the presence of a release trigger (enzyme or(and) pH) at RT was performed by measuring the UV–vis absorption time profile at λ = 280 nm (λ_max_ of EUG) in the supernatant after separation of the solid nanoassemblies by centrifugation over the period of 24 h.

Control experiment series: EUG/CAS-MSNs–COOH (5 mg) were dispersed separately in each of the 1.5 mL portions of 0.1 M PBS buffer adjusted to pH 7.4, 6.0, 1.5, respectively, at RT. The suspension was kept homogeneous by placing it in a rotary mixer throughout the experiment. The suspension was centrifuged (8000 RPM, 8 min) to separate the nanoassemblies at the specified time points (sampling times). A 300 μL portion of the supernatant was collected to monitor the EUG release through absorbance at λ = 280 nm. Separated nanoassemblies were resuspended, and after immediately adding 300 μL of fresh PBS buffer to the suspension, the mixing was continued until the next sampling point. 

Enzyme responsiveness: To quantify the enzyme-responsive release of EUG from EUG/CAS-MSNs–COOH, 5 mg of EUG/CAS-MSNs–COOH was dispersed in 1.1 mL of 0.1 M PBS buffer solution (pH 7.4) followed by addition of 0.4 mL of trypsin solution (0.1 M PBS) of a specified concentration at RT. The sampling of EUG concentrations was conducted as in the control experiment series. Time was set as zero when the trypsin solution was added to the suspension of casein-capped drug-loaded particles. The EUG release profile was observed over 24 h. A series of trypsin-concentrated solutions (400 μL) in 0.1 M PBS buffer (pH 7.4) was used to determine the optimum trypsin concentration, Continuous increasing of trypsin concentration did not substantially increase the EUG release. 

All the EUG release studies were performed in triplicate to obtain standard deviation values for each recorded variable, and Equation (3) determined the cumulative percentage of the released EUG:(3)$$\mathrm{Cumulative}\;\mathrm{percentage}\;\mathrm{release}=\frac{\mathrm{EUG}\;\mathrm{release}\;\mathrm{at}\;\mathrm{time}\;t+\Sigma \;\mathrm{withdrawn}\;\mathrm{EUG}\;\mathrm{before}\;\mathrm{time}\;t}{\mathrm{total}\;\mathrm{EUG}\;\mathrm{loaded}\;\mathrm{into}\;\mathrm{particles}}\;\times \;100$$

### 2.11. Statistical Analysis

All statistical values presented in this study were analyzed with the software GraphPad Prism 9 (GraphPad Software, San Diego, CA, USA) using one-way ANOVA or Student’s t-test. Results are presented as mean ± SEM and α-level were set at 0.05.

## 3. Results

### 3.1. EUG Induces Apoptosis in Colon Cancer Cells

Small molecules with pro-apoptotic activities have shown strong potential value in cancer drug discovery [47], particularly natural alkaloid products [48,49]. EUG-dependent apoptosis has been reported in various human cancer cells, including human colon cancer [14,50,51]. We decided first to confirm whether EUG can induce early and late apoptosis in HCT-116 cells in a time- and dose-dependent manner. The human colorectal carcinoma cell line, HCT-116, was used for the apoptosis assay since this CRC cell line is a highly aggressive cell line with little or no capacity to differentiate, suggesting that HCT-116 cells are predominantly cancer stem cells (CSCs) [52]. Based on previous reports [51], HCT-116 cells were incubated with 125, 250, and 500 μM EUG for 24 h followed by PI/annexin-V flow cytometry analysis. Figure 2A–F show that EUG significantly increased the population of HCT-116 cells in early and late apoptosis. Interestingly, 125 and 250 μM EUG significantly increased the population of cells in the early apoptosis (box Q3 in Figure 2B,C, and blue and green columns in Figure 2E). We saw the highest early apoptosis with 500 μM EUG (purple column in Figure 2E). However, 500 μM EUG significantly pushed HCT-116 cells into late apoptosis (box Q2 in Figure 2D and purple column in Figure 2F). We concluded that EUG induces early and late apoptosis in a dose-dependent manner (Figure 2E,F). The mean percentages of early apoptosis were as follows: DMSO (2.75%), EUG 125 μM (21.7%), EUG 250 μM (31.45%), and EUG 500 μM (50.40%), with a *p*-value of <0.05 to <0.0001 (N = 4). The mean percentages of late apoptosis were as follows: DMSO (1.75%), EUG 125 μM (16.63%), EUG 250 μM (34.8%), and EUG 500 μM (43.82%), with a *p*-value of <0.01 to <0.001 (N = 4). Figure 2A–F indicate that a high dose of EUG is necessary for successful irreversible apoptosis in CRC cells.

In the second set of experiments, HCT-116 cells were subjected to EUG 500 μM treatment for 24, 48, and 72 h followed by PI/annexin-V flow cytometry analysis. Panels G–I in Figure 2 show that EUG significantly increased the population of HCT-116 cells in late apoptosis in a time-dependent manner. The mean percentages of early apoptosis were as follows: EUG 24 h (51.45%), 48 h (28.07%), and 72 h (7.05%), with a *p*-value of <0.0001 (N = 4). The mean percentages of late apoptosis were as follows: EUG 24 h (42.17%), 48 h (67.1%), and 72 h (83.32%), with a *p*-value of <0.0001 (N = 4). As previously reported in breast cancer [14], EUG activates apoptosis in a caspase-3-dependent manner. Figure 2J–M reveal that EUG 500 μM significantly increased the level of caspase-3 in HCT-116 colon cancer cells. The mean percentages of caspase-3 increased from 30.3% in the DMSO (control) group to 89.4% in EUG (500 μM) treated cells with a *p*-value of <0.0001 (N = 4, mean ± standard deviation). Figure 2 indicates that the controlled release of a high dose of EUG is necessary for successful irreversible apoptosis in CRC cells.

### 3.2. EUG Induces Apoptosis in Metastatic Colon Cancer Cells

To validate the apoptotic results achieved in Figure 2 and to determine whether EUG can induce apoptosis in metastatic colon cancer cell lines, the colon cancer cell lines SW480 derived from a primary tumor and SW620 from a metastatic site in the same patient [53] were treated with 125, 250, and 500 μM EUG. After 24 h, cells were subjected to PI/annexin V flow cytometry analysis. Measured early and late apoptosis in Figure 3 show that 500 μM EUG significantly induced late apoptosis in both SW480 and SW620 (purple columns in Figure 3E,F). Interestingly, early apoptosis measured in SW480 cells was not sensitive to EUG in a dose-dependent manner (Figure 3C). Early apoptosis in SW620 cells revealed a significant response to EUG 125 μM and 250 μM (Figure 3D). However, EUG 500 μM shifted cells from early to late apoptosis in SW620 cells (purple columns in Figure 3D,F). The mean percentages of early apoptosis in SW480 were as follows: DMSO (5.95%), EUG 125 μM (24.07%), EUG 250 μM (24.57%), and EUG 500 μM (24.5%), with a *p*-value of <0.0001 (N = 4). The mean percentages of late apoptosis in SW480 were as follows: DMSO (2.6%), EUG 125 μM (9.02%), EUG 250 μM (6.65%), and EUG 500 μM (57.55%), with a *p*-value of <0.05 to <0.0001 (N = 4). The mean percentages of early apoptosis in SW620 are as follows: DMSO (3.18%), EUG 125 μM (59.07%), EUG 250 μM (48.12%), and EUG 500 μM (29.67%), with a *p*-value of <0.0001 (N = 4). The mean percentages of late apoptosis in SW620 were as follows: DMSO (1.18%), EUG 125 μM (25.7%), EUG 250 μM (23.27%), and EUG 500 μM (62.5%), with a *p*-value of <0.0001 (N = 4). 

The results in Figure 3 provide further evidence of the effective apoptotic function of EUG in CRC cells in different cancer stages. This suggests a comprehensive rationale for the selective targeting of metastatic CRC tumors with a high dose of EUG. 

### 3.3. EUG Suppresses Migration and Invasion in the HCT-116 Colon Cancer Cell Line

Migration and invasion are critical steps in cancer metastasis [54]. To investigate whether EUG could suppress CRC cells’ migration and invasion abilities, we used the xCELLigence Real-Time Cell Analysis detection platform to determine the cell adhesion rate as previously described [55] (Figure 4A). HCT-116 cells received 125, 250, or 500 μM EUG and then they were immediately seeded on E-plates (cell adhesion assay) or CIM plates without Matrigel (migration assay) or with Matrigel (invasion assay) as previously described [44,56]. Figure 4B,C shows that EUG significantly reduced the cell adhesion capacity of HCT-116 cells monitored by E-Plates (N = 4, *p* < 0.0001) in a dose-dependent manner. The recorded migration of HCT-116 cells (Figure 4D,E) and analyzed cell index (Figure 4F) show that the 500 μM EUG can significantly decrease cell migration. Similarly, recorded invasion of HCT-116 cells (Figure 4G,H) and analyzed cell index (Figure 4I) show that 500 μM EUG can significantly decrease cell invasion. Results shown in Figure 4 indicate that the release of high concentrations of EUG by NPs at the tumor site can effectively target the biological mechanisms of the metastatic processes developed in human colon cancer cells.

### 3.4. EUG Decreases Population of LGR5^+^, CD44^+^, and CD133^+^ Colon Cancer Stem Cells

It is well-accepted that colon cancer stem cells (CSCs) are the drivers of tumor progression and drug resistance [57]. It has been shown that therapeutic doses of EUG target CSCs in breast cancer by significant downregulation of β-catenin [58,59]. We used flow cytometry to determine whether the EUG can decrease colon CSC populations. We used LGR5, CD44, and CD133 to examine the effect of EUG on CSCs (HCT-116 cells). HCT-116 cells received 125 or 500 μM EUG for 24 h. Panels 5A–E (CD44^+^), G-K (Lgr5^+^), and M-Q (CD133^+^) represent the population of positive cells. Statistical analysis of recorded positive cells (Figure 5F,L,R) revealed that EUG 500 μM significantly decreased the level of receptor-positive CSCs among HCT-116 cells. The mean percentages of the remaining positive CSCs pool in HCT-116 cells treated with 500 μM were as follows: CD44^+^ (18.175%), Lgr5^+^ (1.33%), CD133^+^ (16.125%), with a *p*-value of <0.001 or <0.0001 (N = 4). Both CD44 and LGR5 are involved in local and liver metastasis in colorectal cancer [60]. On the other hand, CD133 CSCs are accepted as a reliable prognostic marker in patients and as a potential target for CRC treatment [61]. In general, colon CSCs are highly tumorigenic, aggressive, and chemoresistant, and they are a critical factor in the metastasis and recurrence of CRC [62]. These data suggest that EUG can potentially suppress the CSCs at higher doses delivered by NPs, increasing the druggability of NPs–EUG as an effective anti-metastatic targeted therapy.

### 3.5. Synthesis and Characterization of Casein-Capped Controlled Eugenol Delivery System

As illustrated in Figure 1, MSN-based nanoassemblies end-capped with casein were successfully fabricated and tested as a controlled eugenol delivery system. The sol-gel reaction method was used to synthesize the base material MSNs. MSNs–NH_2_ were fabricated by grafting APTES on to the surface of MSNs. Subsequently, MSNs–COOH were prepared by introducing succinic anhydride by acyl nucleophilic substitution, as we reported [46]. To covalently couple the casein to the MSNs–COOH, EDC, and sulfo-NHS were added to the MSNs–COOH suspension. Then, casein was added to the suspension, and the mixture was stirred for 5 h to couple the amine group of casein to the carboxyl group of nanoassemblies. The synthesized nanoassemblies were characterized by transmission electron microscopy (TEM), scanning electron microscopy (SEM), powder X-ray diffraction spectra (PXRD), Fourier transform infrared spectroscopy (FT-IR), and dynamic light scattering (DLS).

The morphology and size of the nanoassemblies were evaluated using SEM and TEM imaging (Figure 6A). The synthesized MSN particles had a uniformly spherical shape with an average diameter of 95 nm. After covalently coupling with casein, the average diameter of nanoassemblies (EUG/CAS-MSN–COOH) increased to about 115 nm while exhibiting almost unchanged morphology (Figure 6A(a,b)). The well-ordered hexagonal mesopore structures were revealed on the TEM images for nanoassemblies (Figure 6A(c)). The small-angle powder X-ray diffraction patterns (PXRD) for nanoassemblies further illustrated the formation of a well-ordered hexagonal structure from the diffraction peaks within the ranges of 2.3° to 2.35° and 4.0° to 5.0° (2θ), which can be assigned to the (100), (110), and (200) diffraction planes of the unit cell. There was a decrease in peak intensity for EUG/MSNs–COOH, showing a slight distortion of the ordered mesopore structure (Figure 6B). After coupling with casein, the XRD spectrum peaks for EUG/CAS-MSNs–COOH were much weaker, indicating that the pores of nanoassemblies were capped by casein, which is further confirmed by the blurred pore structure from TEM images of EUG/CAS-MSNs–COOH (Figure 6A(d)).

The hydrodynamic diameter, size distribution (PDI), and surface charge for nanoassemblies were evaluated by DLS analysis and zeta potential measurements. As presented in Figure 6C, the hydrodynamic diameter of MSNs, EUG/MSNs–COOH, and EUG/CAS-MSNs–COOH (optimized casein content) was approximately 191 nm (PDI-0.287), 210 nm (PDI-0.379), and 245 nm (PDI-0.313), respectively. Due to the solvation layer at the particle surface in the solution subjected to the DLS analysis, hydrodynamic diameter values were larger than the particle diameter exhibited in the TEM images. An increase of around 35 nm in average diameter for casein-capped particles after casein conjugation confirms successful casein coupling to the particle surface. The zeta potentials of MSNs, EUG/MSNs–COOH, and EUG/CAS/MSNs–COOH were −26.33 mV, −48.65 mV, and −58.54 mV, respectively (Figure 6D). The silanol groups on the MSN’s surface account for the negative charge of the surface. The nanoassemblies gained more negatively charged groups after introduction of carboxy-groups and coupling with casein [63], which was reflected in their zeta-potentials.

Surface-incorporated layers and the presence of eugenol in MSN particles were confirmed by the FTIR spectra for MSNs, EUG/MSNs–COOH, and EUG/CAS-MSNs–COOH (Figure 6E). All FTIR spectra showed similar characteristic bands of mesoporous silica at 1070 cm^−1^ (asymmetric Si-O-Si stretching), at 799 cm^−1^ (symmetric Si-O stretching), 3400 cm^−1^ (OH), and 965 cm^−1^ (Si-OH stretching) [64,65]. After grafting COOH groups to MSNs, one of the new bands appeared at 1670 cm^−1^ belonging to the C=O vibration of the COOH group, which confirmed the successful grafting (Figure 6E(b,c) Characteristic bands of eugenol were also located at (2990, 2930) cm^−1^ (CH stretching), 1635, 1590, and 1510 cm^−1^ (C=C aromatic ring) [66,67]. After eugenol encapsulation, both EUG/MSNs–COOH and EUG/CAS/MSNs–COOH nanoassembly spectra showed new characteristic peaks of eugenol, which confirmed that the eugenol was successfully encapsulated into the MSNs (Figure 6E(b,c)). The intensity of the C-H stretching band between 2900 and 3100 cm^−1^ increased considerably for EUG-loaded particles, confirming successful eugenol encapsulation into the MSNs [66]. The presence of casein in the casein-capped MSNs’ surface was also confirmed by FTIR spectroscopy (Figure 6E(c)). The presence of new peaks between 1650 cm^−1^ and 1500 cm^−1^ for EUG/CAS-MSNs–COOH, which can be attributed to the characteristic of amide I and amide II, are present in the FTIR spectra of the synthesized casein-capped MSNs [68]. 

### 3.6. Eugenol (EUG) Loading, Casein Coupling and Enzyme Responsive In Vitro Release of Eugenol

A schematic diagram of EUG loading into nanocarriers is shown in Figure 7A. To find out the optimal eugenol amount that can be loaded into the nanoassemblies with high drug loading amount and efficiency, a series of different eugenol (EUG) amounts containing 150, 75, 37.5, 30, 15, and 7.5 µg was loaded into 5 mg of nanoassemblies. The calculated drug loading efficiency (DLE) and encapsulation efficiency (DEE) are presented in Figure 7B. The DLE was proportionate to the EUG concentration, while the DEE decreased with the increased EUG concentration. The optimal EUG concentration was 25 µg/mL, where DEE was high (77.2%) and DLE was 0.579%.

After obtaining the optimal EUG amount contained in the MSN nanoassemblies, we determined the optimum amount of casein required to achieve maximum EUG encapsulation inside the MSN nanoassemblies (Figure 8A). As shown in Figure 8B, casein at given concentrations (0.1%, 0.2%, 0.3%, 0.4%, 0.5%, 0.6%, 0.7%, and 0.8%, *w*/*v*) was covalently conjugated to the MSN nanoassemblies, followed by the drug-releasing experiments to determine the optimal amount of casein that can hold the maximum amount of loaded EUG inside the MSN channels for 24 h. As depicted in Figure 8C, there was no significant EUG encapsulation for 24 h in the control experiment (0% casein *w*/*v*), indicating that without the pore-blocking agent of casein, eugenol was immediately released from the open pore channels of MSN nanoassemblies without substantial retention. As the amount of the attached casein increased, there was a gradual improvement in the EUG encapsulation efficiency until casein *w*/*v*% reached 0.6% *w*/*v*. As shown in Figure 8C, 73.8% of loaded EUG was held for 24 h inside the MSN nanoassemblies after capping the pore channels with 0.6 *w*/*v*% casein. Further increase in the amount of conjugated casein resulted in a marginal increase in EUG encapsulation efficiency, suggesting that 0.6% casein *w*/*v* iss the optimal value that efficiently holds EUG for 24 h inside the MSN nanoassemblies. These observations informed our selection of 0.6% *w*/*v* casein for synthesizing EUG/CAS-MSNs–COOH nanoassemblies for all EUG release experiments. The amount of casein conjugated to MSN nanoassemblies was determined by the Bicinchoninic acid (BCA) protein assay. In this assay, after casein conjugation, the remaining free casein in the MSN nanoassemblies dispersion was separated by centrifugation and analyzed by the BCA assay. According to the BCA assay, covalent conjugation of casein at its optimal concentration to the MSN nanoassemblies exhibited a coupling efficiency of 28.9%.

After identifying the optimum casein *w*/*v*% for the substantial retention of EUG inside the MSN nanoassemblies, we performed drug release experiments at different pH values and demonstrated that the EUG/CAS-MSNs–COOH can hold the drug inside the MSNs channels at a wide pH-range (1.5–7.4) for 24 h in PBS. Three different pH conditions were selected to simulate the conditions of the intestine (pH 7.4) [69], colon (pH 6.0) [70], and stomach (pH 1.5) [71], and EUG release by EUG/CAS-MSNs–COOH in the absence of any enzymes at RT (Figure 8D). Therefore, casein maintains the pores sealed over the pH range of 1.5–7.4 in the absence of any enzyme.

Figure 9A illustrates the enzyme-responsive drug release experiments with EUG/CAS-MSNs–COOH. Trypsin solutions (0.4 mL) at 0%, 0.01%, 0.05%, 0.1%, 0.2%, and 0.25% *w*/*v*% concentrations were added separately to each of the 1.1 mL portions of the PBS buffer (pH 7.4, RT), containing 5 mg of EUG/CAS-MSNs–COOH (0.6 *w*/*v*% casein) to determine the optimum concentration of trypsin that triggers significant EUG release from EUG/CAS-MSNs–COOH (Figure 9B). The time zero was set when the enzyme was added to the suspensions of the nanoassembly, and the release of EUG was monitored for 24 h (Figure 9C). In the control experiment (without trypsin), only ~25% of EUG leaked over 24 h from the nanoassembly. In the presence of trypsin, the cumulative release of EUG for 24 h at pH 7.4 was about 49%, 65%, 71%, 80%, and 79%, at 0.01%, 0.05%, 0.1%, 0.2%, and 0.25% *w*/*v*% trypsin concentrations, respectively (Figure 9C). The time release profiles demonstrate that trypsin triggered the release of EUG in a concentration-dependent manner until its concentration of 0.2 *w*/*v*%. Further increase in the trypsin concentration did not result in a more efficient release of EUG from the pore channels (Figure 9C). Therefore, 0.2 *w*/*v*% is the optimum trypsin concentration that triggers ~80% release of EUG for 24 h from the EUG/CAS-MSNs–COOH at pH 7.4, RT (Figure 9D). The UV absorption spectra of pure EUG and the EUG released into the release medium confirmed the chemical stability of EUG during nanoparticle drug loading and release (Appendix A).

### 3.7. Casein-Coated NPs–EUG Decreases Cell Viability in Cancer Cells While Normal Colon Epithelial Cells Remain Intact

Active matrix metalloproteinases (MMPs) are predominantly released at the leading edges of migrating tumor cells [39], and they play an important role in the development and progression of CRCs [72,73]. Several MMPs, including MMP-7, are overexpressed in human CRCs [74]. MMP-7 is substantially expressed by CRCs [75], and its concentration increases in a stage-dependent manner in CRCs [76]. In fact, MMP-7 has an approximately six-fold greater expression in tumor masses versus normal cells [77]. Therefore, MMP-7 can potentially be utilized for the site-specific release of anti-metastatic drugs against human CRC primary and metastatic tumors. Based on the current reports, we conducted a set of Western blots (WBs) to confirm that only colon cancer cells contain a large pool of MMP-7 (Figure 10A). HCT-116, HT-29 (a human-derived colorectal adenocarcinoma), FHC (a normal colon epithelial cell), and HEK 293 cell lines (widely known as the Human Embryonic Kidney 293 cells) were subjected to WBs using an anti-MMP-7 antibody (Appendix A). HT-29 colon cancer cells exhibit an enhanced metastatic potential and therefore were used as a positive control next to HCT-116 and FHC cells. WB results revealed that HCT-116 and HT-29 contain an enriched pool of MMP-7 in the cytoplasm, while FHC showed no detectable intracellular MMP-7 (Figure 10A and Appendix A). Interestingly, HEK293 cells showed a trace amount of MMP-7 protein in the cytoplasm compartment (Appendix A). Based on WB results, FHC normal colon epithelial cells and HCT-116 colon cancer cells were treated with 125 μM pure EUG or 125 μM EUG delivered by casein-coated NPs (Figure 10B). After 48 h, cells were subjected to a crystal violet cell cytotoxicity assay (Figure 10C,E). Appendix A shows validation of the cytotoxicity assay. The flow chart in Appendix A illustrates the steps of the crystal violet cell cytotoxicity assay. Figure 10 shows that 125 μM pure EUG had no effect on FHC cells but did significantly decrease the viability of HCT-116 cells (red columns versus blue columns in panels D and F in Figure 10). The same concentration of EUG delivered by casein-coated NPs further decreased cell viability in HCT-116 cancer cells (Figure 10F, green column). Casein-coated NPs with and without EUG in FHC normal cells (Figure 10D, green and purple columns), and casein-coated NPs without EUG in cancer cells (Figure 10F, purple column), showed an elevation of adherent cells due to the extra amount of available casein. As previously reported, casein increases cellular adhesion through the E-cadherin/beta-catenin mechanism [78]. To summarize, the order of toxicity in HCT-116-cancer cells was as follows: NPs–EUG > pure EUG > control group > Casein-coated NPs without EUG. Based on the WB results (Figure 10A), we decided to examine the effect of pure EUG and casein-coated NPs with EUG on HEK-293 cells. Appendix A shows that pure EUG had no effect on HEK-293 cells compared to DMSO (0.1% *v*/*v*) control group. Interestingly, EUG delivered with casein-coated NPs significantly reduced cell viability in HEK-293 cells (green column). As observed in HCT-116 and FHC cells (Figure 10), casein-coated NPs without EUG significantly increased cell adhesion in HEK-293 cells (Appendix A). Overall, the casein-coated NPs–EUG was able to successfully deliver its toxicity to cancer cells while it failed to similarly perform in normal cells. Therefore, EUG is the dominant player in terms of cell toxicity when we used casein-coated NPs–EUG on cancer cells. We believe the improvement of cell viability in normal cells and cancer cells (purple column) by casein is heavily mediated by the effect of casein on cell adhesion [78], although we cannot eliminate the casein’s crosstalks with other cell component or reagents in the assay kit. Further studies using different cell toxicity assay will verify achieved results illustrated in Figure 10. 

In the second set of experiments, HCT-116 and FHC cells were treated with casein-coated MSNs–FITC for 90 min, followed by confocal microscopy as previously described [46]. The results in Figure 10G,H indicate that HCT-116 colon cancer cells are associated with a substantial portion of NPs bound to the plasma membrane. We observed no association with, or internalization of, the negatively charged MSN–FITC NPs in normal colon epithelial cell culture (Figure 10H). The Z-stack pictures in panels G–H (Figure 10) indicate a percentage of NPs–FITC internalized into the cytoplasm compartment of HCT-116 cells as previously described [79]. Results in Figure 10 suggest that the engineered casein-coated NPs in this study can selectively deliver and release an effective dose of their cargo (EUG) at tumor sites. At the same time, their low-affinity to normal cells with low or no active MMP-7 enzymes remain dominantly unexposed to high local concentrations of drugs.

## 4. Discussion

CRC is the third leading cause of cancer death in the USA, with an estimated 52,000 deaths in 2022 [80]. Dissemination of the primary tumor to distant sites such as the liver and lungs is the primary cause of death in the majority of patients [81]. With limited therapeutic options for advanced CRC, there is a clear need to develop more effective targeted therapies to decrease the high mortality rates associated with metastatic disease [82]. According to current evidence, suppression of pathways involved in cell migration/invasion fed by epithelial–mesenchymal transition and inhibition of cancer stem cells can effectively slow down tumor metastasis in a broad range of cancer types, including CRC [62,83]. However, despite the strong therapeutic potential of these tumorigenic targets, there are currently no effective targeted therapies. This study has explored the potential of a natural anticancer molecule, eugenol, for CRC therapeutics using a novel nanotechnology-based delivery system and underlying mechanisms of its action. Adequate drug concentrations of EUG delivered by casein-coated nanoparticles at the tumor site is expected to effectively block severe undesirable side effects and decrease the incidence of multiple drug resistance. Ultimately, this study provides a platform for a new generation of targeted therapies for patients with metastatic cancers.

To further understand the anticancer mechanism of EUG against metastatic human colon cancer, cancer cells in different metastatic stages were treated with EUG. As the apoptotic effect of EUG in diverse malignant tumors has previously been shown [50,84,85], we demonstrated that EUG can induce early and late apoptosis in human colon cancer cells in a dose- and time-dependent manner and significantly decrease cell migration and invasion, the two characteristic features used by metastatic cancer cells to invade adjacent tissues and metastasize to other organs [86]. We observed a similar inhibition in cell migration and invasion in both the presence and absence of FBS (starved versus non-starved conditions). Our results indicate that EUG interferes with major tumorigenic pathways involved in tumor metastasis, such as the KRAS pathway [87,88,89]. Published data suggest that eugenol alone, or its combination with cisplatin, can target CSCs in breast and ovarian cancers, respectively [58,59,85]. Our flow cytometry experiments revealed that a high dose of EUG (500 μM) effectively decreases the population of CD44+, CD133+, and LgR5+ cells in a metastatic colon cancer cell line. It has been shown that CD44 plays a key role in colon cancer invasion [90]. CD133 and Lgr5 cells are independent prognostic markers for low survival and drug resistance in colon human colorectal cancer patients [91,92]. The anti-CSC’s effect of a high dose of EUG further speaks to the therapeutic potential of casein-coated nanoparticles carrying high doses of EUG while minimizing its adverse effect on normal cells. Further studies will determine the underlying anticancer mechanism of EUG in CRC animal models using casein-coated NPs. 

We engineered casein-coated mesoporous silica nanoparticles (EUG/CAS-MSNs–COOH) that efficiently encapsulate EUG, an anticancer drug molecule inside the MSN pore channels. We determined the optimal concentration of EUG to achieve efficient drug encapsulation efficiency and loading capacity by the MSN nanoassembly. We determined the optimum casein concentration that can efficiently trap the drug inside the particles. The release of EUG drug molecules from EUG/CAS-MSNs–COOH via enzymatic cleavage of the casein gatekeeping element (MMP-7 substrate) with trypsin proved the concept of the enzyme-triggered controlled release of EUG. Without a release-triggering enzyme, the EUG-loaded casein-capped MSNs did not exhibit any undesired EUG leakage at a wide range of pH values of the GI tract. 

It is well established that MMPs play an important role in the development and progression of CRC [72,73]. MMP-1, -2, -3, -7, -9, -13, and MT1-MMP are overexpressed in human CRC [74], and their expression levels are associated with poor prognosis in CRC patients. Interestingly, EUG decreases the level of MMPs released by cancer cells [93] which indicates MMP-7-triggered release of eugenol can be a self-regulated mechanism. This later effect can turn into a valuable bioactivatable probe specific for MMP activity by measuring MMP activity in CRC animal models treated with NPs–EUG versus typical chemotherapeutic drugs. Further, down-regulation of MMP-7 expression by EUG may contribute to its suppressing effect on the migration and invasion of cancer cells, decreasing their aggressiveness. Determining NPs–EUG’s pharmacodynamics in animal models is a critical step to verifying the safety and efficacy of NPs–EUG. An ongoing project in our group focuses on the PK/PD of NPs–EUG in two mouse models of colorectal cancer. Future animal models will illustrate whether the NPs–EUG can dominantly function against metastatic tumors. Besides NPs–EUG treatment, animals with CRC (primary and metastatic models) will be treated with the combination therapy of NPs–EUG and traditional chemotherapies to determine if we can observe a synergistic effect. A synergistic effect of NPs–EUG combined with a chemotherapeutic agent can decrease the dosage of chemotherapeutic agents used in patients and lessen the incidence of drug resistance. 

## 5. Conclusions

We found that free EUG has no cytotoxic effect on either normal epithelial colon cells or on HEK-293 cells. In contrast, free EUG induced cytotoxicity in HCT-116 cells. Even though EUG-loaded NPs contain the same drug concentration as the free EUG, there are several advantages for using NPs as a potential CRC therapeutic. NPs can release the loaded EUG directly at the tumor site allowing for a higher concentration of drug which enhances the cytotoxic effect. This specificity allows for reduced toxicity in the surrounding normal healthy tissue experienced by using free EUG [94]. The traces of MMP-7 in HEK-293 cells and its tumorigenic characters [95] may account for the cytotoxicity of casein-coated and EUG-loaded NPs for those cells, which speaks to the expected role of MMP-7 in the release of EUG. The particles were not cytotoxic toward the normal FHC cell line. The enhanced effect of NPs delivering drugs can be due to the different mode of cell penetration and consequent higher internalization resulting in high local concentrations of the drug at the tumor site as previously described [96,97]. Our confocal microscopy studies confirmed that casein-coated NPs labeled with a fluorescent dye (FITC) had no affinity to FHC cells while they showed a high affinity and internalization in HCT-colon cancer cells. 

Mechanistic studies of EUG delivered by an innovative use of nanotechnology for gated drug delivery will facilitate the development and clinical translation of NPs+EUG to new targeted therapies against primary and metastatic tumors. Based on the structure and composition of casein-coated NPs and the low level of MMP-7 in plasma compared to the tumor sites, the intravenous (IV) injection of NPs–EUG will be the first choice in the pre-clinical model. Most current chemotherapeutic agents used in CRC patients, such as 5-FU (5-fluorouracil), are typically given as an injection into a vein (IV). Certainly, future modifications of NPs–EUG, such as replacement of casein with fibronectin, a known in vivo MMP-7 substrate [98], can open other administration routes besides IV, such as direct injection into the tumor mass under colonoscopy guides. According to a published assessment [99], we expect diverse therapeutic effects using different routes in the animal models. 

## Figures and Tables

**Figure 1 cancers-15-01145-f001:**
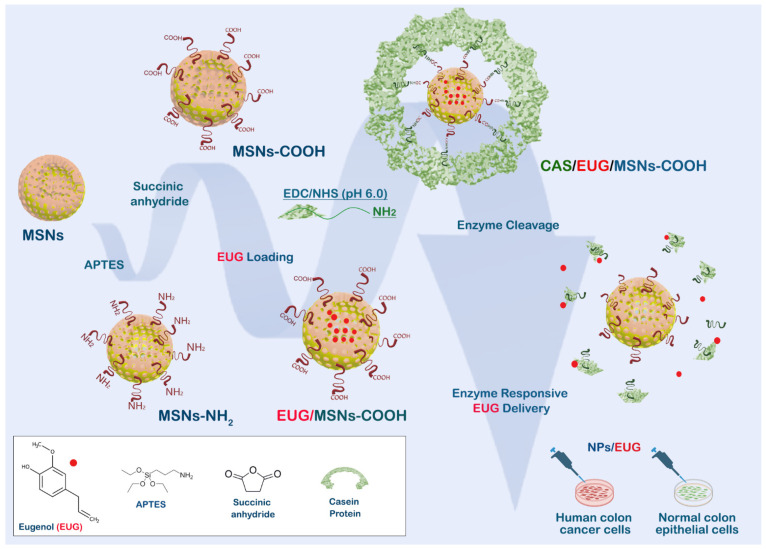
Design and functioning of the nanoassembly for targeting colon cancer cells. MSN is modified by carboxy groups, loaded with EUG, and covalently sealed by casein. Digestion of casein by MMPs released by cancer cells maximizes the toxicity of EUG against cancer cells while protecting healthy cells.

**Figure 2 cancers-15-01145-f002:**
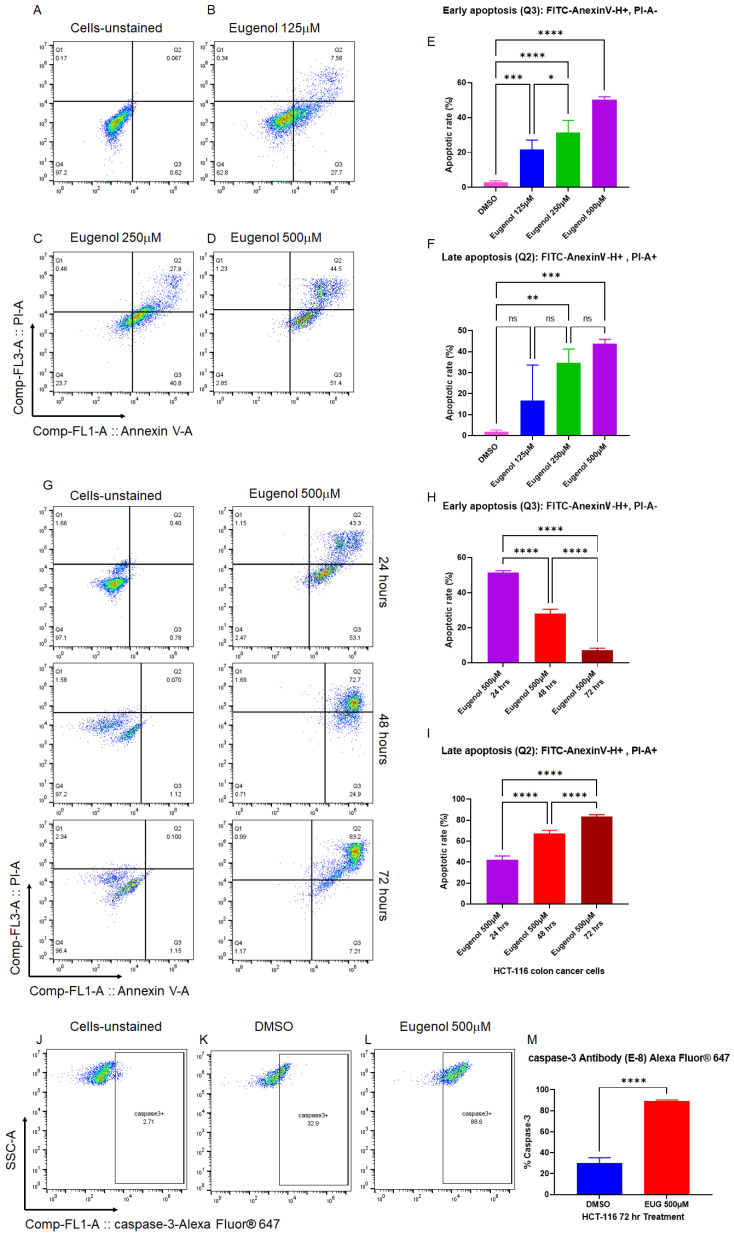
EUG induces apoptosis in colon cancer cells in a dose- and time-dependent manner. Panels (**A**–**F**): HCT-116 colon cancer cells were incubated with DMSO or EUG (125, 250, and 500 μM) for 24 h, followed by PI/annexin V flow cytometry analysis. Panels (**G**–**I**): HCT-116 cells were subjected to EUG 500 μM treatment for 24, 48, and 72 h followed by PI/annexin V flow cytometry analysis. Collected flow cytometry results were analyzed with FlowJo software to determine the % of Annexin V-positive (early apoptosis) versus Annexin V/PI-positive cell populations in the presence and absence of EUG. Annexin V-positive—PI-negative populations represent cells with an intact membrane (early apoptosis), while Annexin V-positive—PI-positive staining indicates cells with compromised membranes (late apoptosis). Cells in late apoptosis act like necrotic cells. Graphical representation of FlowJo results in early and late apoptosis performed and expressed as the mean ± standard deviation. Panels (**J**–**M**): Cancer cells positively stained with caspase-3 Antibody-Alexa Fluor 647 show EUG significantly increases caspase-3, a principal marker of apoptosis. Four independent experiments were carried out in triplicate (*p*-value * < 0.05, ** < 0.01, *** < 0.001, **** < 0.0001). ns—non significant.

**Figure 3 cancers-15-01145-f003:**
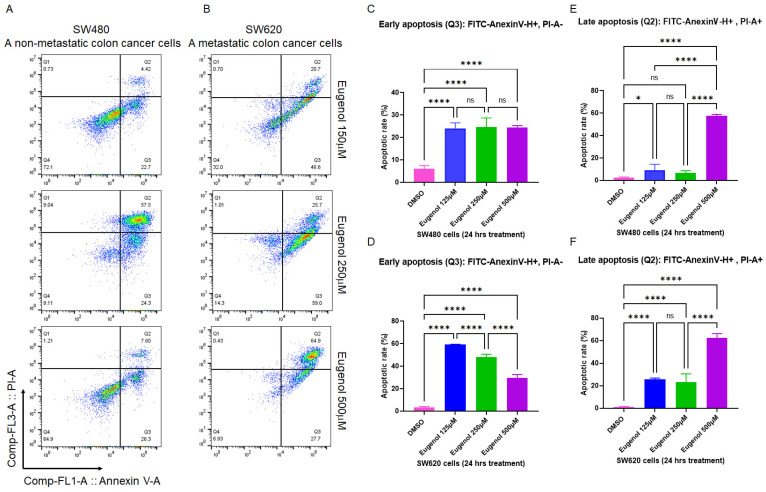
EUG induces apoptosis in metastatic colon cancer cells. Panels (**A**–**F**): The two isogenic human colon cancer SW480 and SW620 cell lines belong to separated tumor masses removed from the same patient. SW480 cells are initiated from the primary tumor, whereas the SW620 cells form the metastatic lymph node lesion. SW480 (Primary CRC) cells and SW620 (metastatic CRC) cells were incubated with DMSO or EUG (125, 250, and 500 μM) for 24 h, followed by PI/annexin V flow cytometry analysis. Collected flow cytometry results were analyzed using FlowJo software to determine the % of Annexin V-positive (early apoptosis) versus Annexin V/PI-positive cell populations in the presence and the absence of EUG. Panels (**A**,**C**,**E**) show significant induction of late apoptosis in SW480 by 500 μM EUG. Similar significant late apoptosis signals were measured in metastatic SW620 cells in the presence of 500 μM EUG (Panel (**F**)). Meanwhile, measurement of early apoptosis (Annexin-positive, PI-negative) showed a sensitivity of both cell lines to EUG (Panels (**C**,**D**)). Four independent experiments were carried out in triplicate (*p*-value * < 0.05, **** < 0.0001, mean ± standard deviation). ns—non significant.

**Figure 4 cancers-15-01145-f004:**
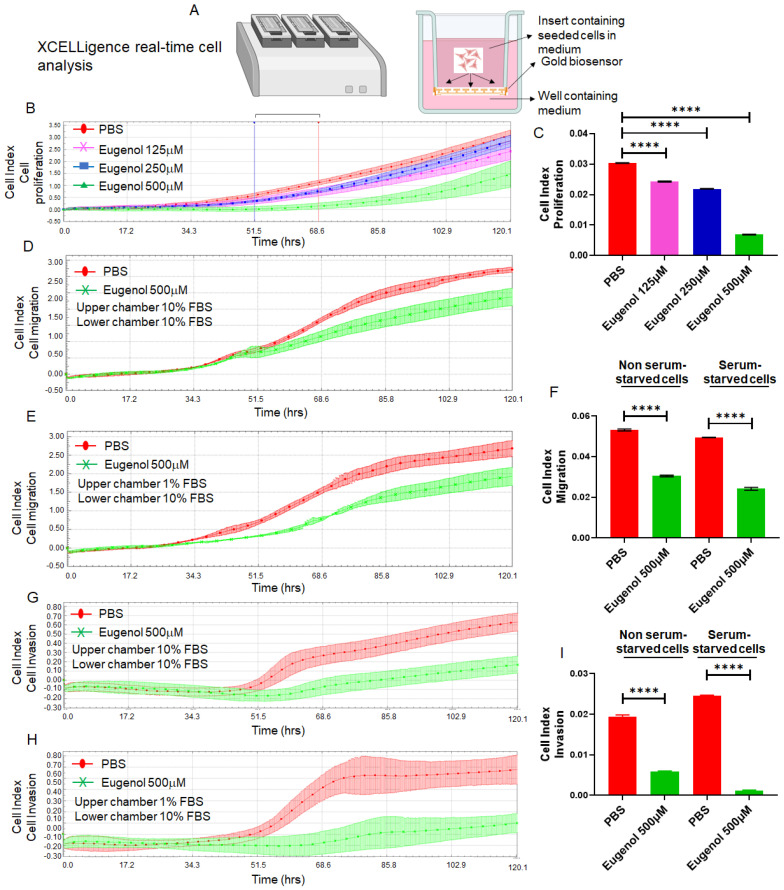
EUG significantly reduces cell adhesion, cell migration, and invasion in human colon cancer cells. Panel (**A**): The xCELLigence RTCA technology is an accurate platform for non-invasive measurement of cell adhesion/viability and migration/invasion in live cells. HCT-116 cells were monitored in real-time for cell adhesion, migration, and invasion for 120 h in the presence of EUG. Graphs (panels (**B**,**D**,**E**,**G**,**H**)) and statistical analysis of calculated slopes (panels (**C**,**F**,**I**)) using critical time points between 50 h and 70 h post-treatment (Blue and red lines in panel (**B**)) show that EUG significantly suppressed cancer cell adhesion, migration, and invasion. Experiments carried out in both the absence and presence of serum starvation (Panels (**F**,**I**)) had no effect on EUG’s suppressive effects, indicating that EUG targets key signaling pathways involved in migration and invasion. The blue (50 h timeline) and red (70 h timeline) in panel (**B**) show critical time points used for analyzing the cell indexes recorded in adhesion, migration, and invasion assays in live cells. Experiments were repeated two times with N of 4 per cell line per experiment (**** *p* < 0.0001, mean ± SD).

**Figure 5 cancers-15-01145-f005:**
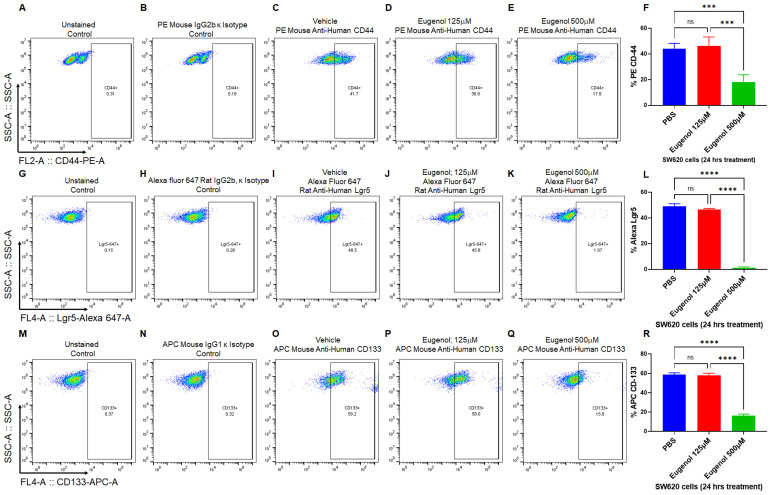
EUG decreases cancer stem cell populations in human colon cancer cells. CSCs positive for Lgr5, CD44, and CD133 cell populations are associated with poor prognosis and drug resistance in CRC tumors. HCT-116 colon cancer cells were treated with DMSO or EUG (125 and 500 μM). After 24 h, cells were subjected to flow cytometry analysis using three CSC markers (CD44, Lgr5, and CD133). Cells treated with 500 μM EUG showed a significant reduction in CD44^+^ (**A**–**F**), LGR5^+^ (**G**–**L**), and CD133^+^ (**M**–**R**) (N = 4, *** *p* < 0.001, **** *p* < 0.0001, mean ± SD). Unstained cells and Mouse IgG2b or IgG1 Isotype controls (panels (**B**,**H**,**N**)) were used to confirm the accuracy of measurements with antibodies. ns—non significant.

**Figure 6 cancers-15-01145-f006:**
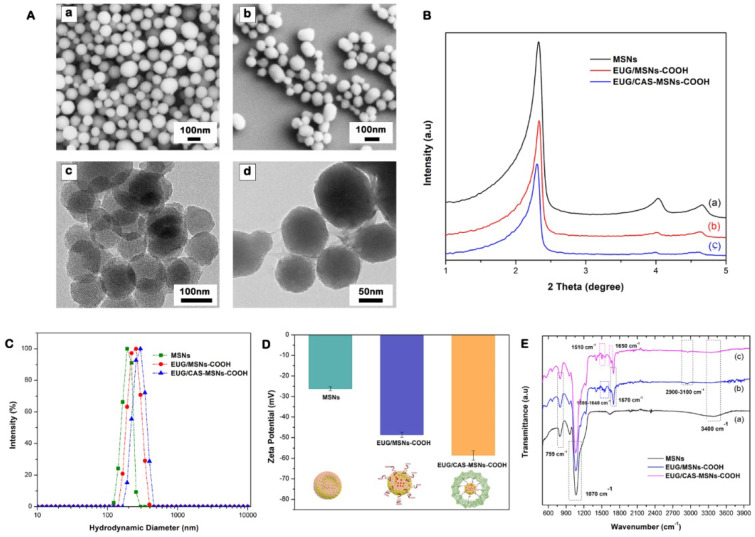
Synthesis and characterization of casein-capped controlled eugenol delivery system (**A**) SEM and TEM images, subpanels (**a**,**b**) illustrate SEM images of MSNs and EUG/CAS-MSNs–COOH, respectively. Subpanels (**c**,**d**) illustrate TEM images of MSNs and EUG/CAS-MSNs–COOH, respectively. (**B**) XRD patterns, (**C**) zeta size distribution, (**D**) zeta potential at pH 7.4, (**E**) FTIR spectra of for MSNs, EUG/MSNs–COOH, and EUG/CAS-MSNs–COOH.

**Figure 7 cancers-15-01145-f007:**
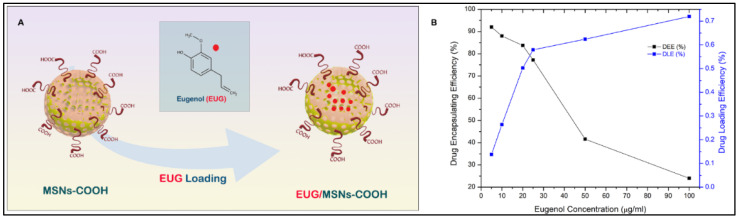
(**A**) Eugenol EUG loading into the MSN pore channels, (**B**) Effect of EUG concentration on drug loading efficiency (DLE) and drug encapsulation efficiency (DEE) of MSN nanoassemblies, 25 µg/mL EUG concentration was identified as the optimum concentration that was efficiently loaded to the MSN channels.

**Figure 8 cancers-15-01145-f008:**
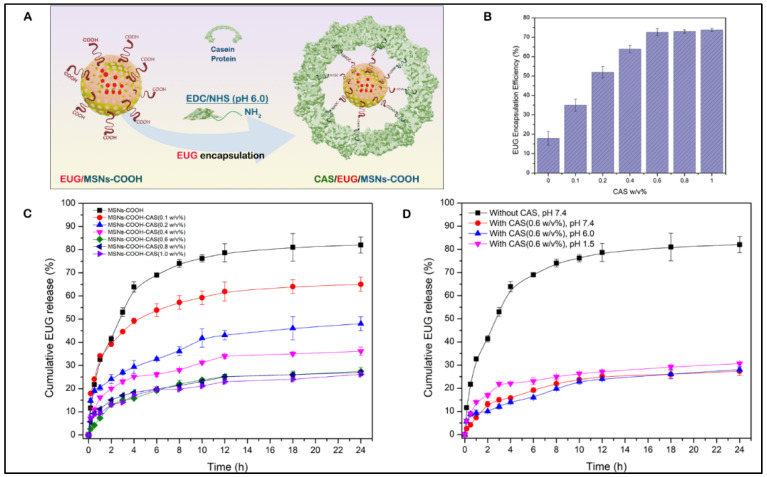
(**A**) Conjugation of casein to the carboxy-functionalized MSN particles after NHS/EDC activation, (**B**) The encapsulation efficiency of eugenol (EUG) for 24 h along with varying *w*/*v*% of casein as a capping layer for EUG-loaded MSN nanoassemblies, (**C**) Cumulative release (%) profiles of EUG from EUG/CAS-MSNs–COOH at different *w*/*v*% of casein at pH 7.4, RT, to obtain the effect of casein capping layer for EUG release over 24 h. The black curve represents EUG release profiles without any casein conjugation to the EUG/MSNs–COOH. (**D**) Cumulative release (%) profiles of EUG from EUG/CAS-MSNs–COOH (0.6 *w*/*v*% casein) at different pH conditions, pH 7.4 (Red curve), pH 6.0 (Blue curve), pH 1.5 (Pink curve). The black curve represents EUG release profiles without any casein conjugation to the EUG/MSNs–COOH at pH 7.4, RT. Each data point represents the mean of the experiments performed in triplicate and their corresponding standard deviation.

**Figure 9 cancers-15-01145-f009:**
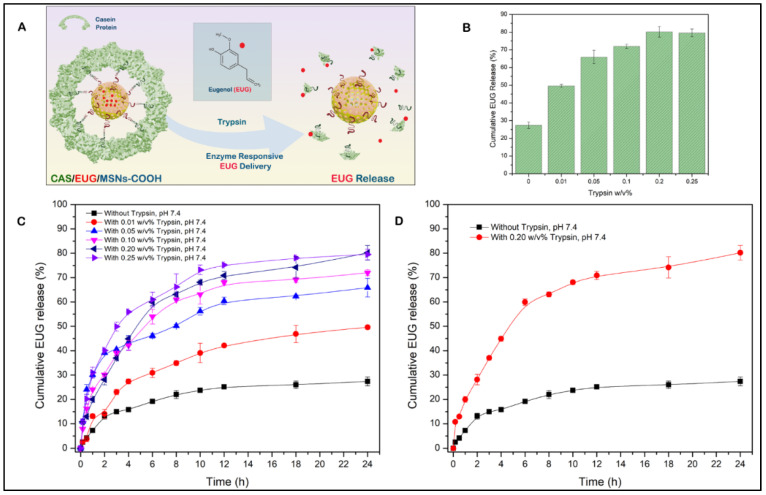
(**A**) Release of an anticancer drug, EUG, which is triggered by trypsin that cleaves a gatekeeping component (casein) of the EUG/CAS-MSNs–COOH, (**B**) The cumulative release of EUG for 24 h at pH 7.4, RT at different *w*/*v*% of trypsin from the MSN–COOH pore channels, (**C**) Cumulative release (%) profiles of EUG from EUG/CAS-MSNs–COOH with and without trypsin at different *w*/*v*% (pH 7.4, RT) for 24 h. The black curve represents EUG release profiles from the EUG/MSNs–COOH in the absence of trypsin at pH 7.4, RT, (**D**) Cumulative release (%) profiles of EUG from EUG/CAS-MSNs–COOH in the presence of the 0.20 *w*/*v*% trypsin mixture (optimized concentration) (red curve) and the absence of any enzymes from casein-capped MSNs–COOH (black curve) at pH 7.4, RT. Each data point represents the mean of the experiments performed in triplicate and their corresponding standard deviation.

**Figure 10 cancers-15-01145-f010:**
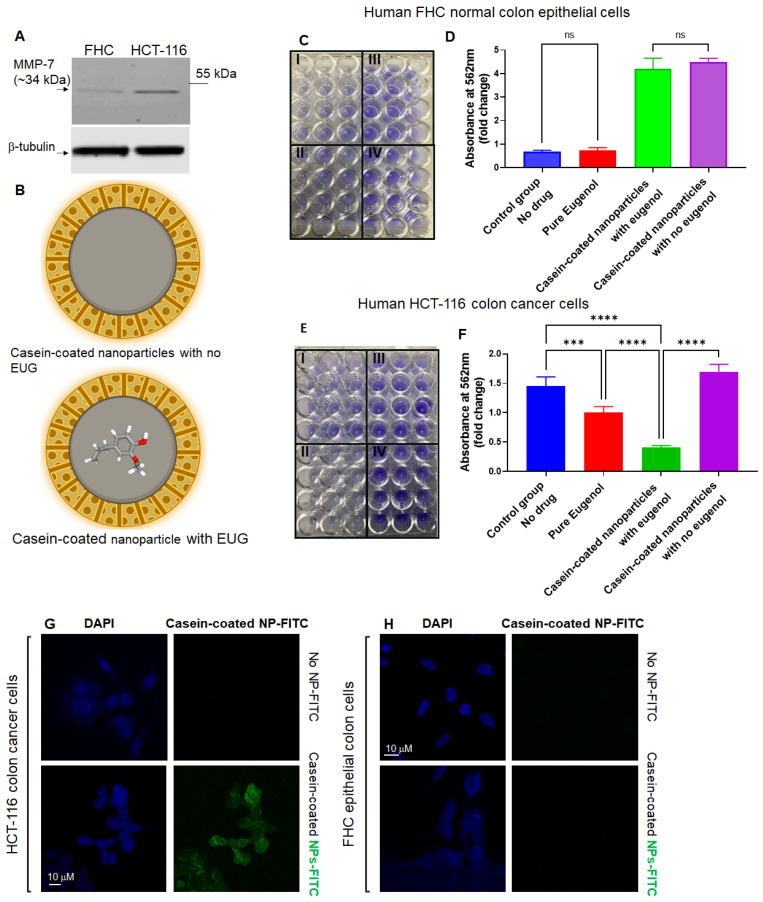
EUG delivered by casein-coated NP decreases cell viability in HCT-116 cells while FHC cells remain intact. (**A**) Anti-MMP-7 shows high expression of MMP-7 in colon HCT-116 colon cancer cell line, while FHC cells showed no trace of intracellular MMP-7. The whole western bolts is shown in Appendix A. HCT-116 and FHC cells were treated with pure EUG (125 μM), casein-coated NPs without EUG, and casein-coated NPs with 125 μM EUG (panel (**B**)) for 48 h, followed by crystal cell viability assay. Absorption of stained cells per well was measured by plate reader. Panels (**C**,**E**) show the attached FHC and HCT-116 cells stained with crystal violet. I. Pure Eugenol, II. Casein-coated nanoparticles with EUG. III. Casein-coated nanoparticles with no eugenol, and IV. control group (DMSO). Calculated results show that pure EUG significantly decreased cell viability in HCT-116 cells, and had no effect on FHC cells (red column versus blue column in panels (**D**,**F**)). More importantly, delivery of 125 μM EUG by NPs to HCT-116 cells further enhanced the reduction in cell viability (green column in panel (**F**)). The casein-coated NPs without EUG increased the adherence of cells in HCT-116 and FHC (N = 4, *p* *** < 0.001, **** < 0.0001, mean ± SD). Panels (**G**,**H**). HCT-116 and FHC were incubated with casein-coated MSNs–FITS (green signals) particles for 90 min and stained with DAPI (blue signals). The Z-stack images show that only HCT-116 colon cancer cells have an affinity to MSNs, and 90 min was enough for partial internalization of NPs to the cytoplasmic compartment (Scale bar is 10 μM). ns—non significant.

## Data Availability

The data presented in this study are available in this article (and Appendix A).

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
