# Peer review of "Enhancing Anti-Tumorigenic Efficacy of Eugenol in Human Colon Cancer Cells Using Enzyme-Responsive Nanoparticles"

_cancers, 2023, doi:10.3390/cancers15041145_

Round 1

Reviewer 1 Report

The authors (Wijewantha et al.,) reported the manuscript entitled Enhancing Anti-tumorigenic Efficacy of Eugenol in Human Colon Cancer Cells Using Enzyme-Responsive Nanoparticles. Several natural plant-based products, and their active secondary metabolites, have been successfully employed as anticancer compounds as effectively target cancer cells by different signaling pathways and cellular mechanisms. While selective delivery and release of these plant-based anti-cancer compounds are still challenging. Eugenol (EUG), is one of the plant-based anti-cancer compounds which demonstrated diverse apoptotic and anti-growth properties against malignant tumors, including colorectal cancer cells. However, the rapid metabolism, excretion, and side effects on normal cells at higher dosages limit its therapeutic application in the treatment of colon cancer. In quest of that authors successfully fabricated EUG-loaded enzyme-responsive nanoparticle (eNPs) and their high affinity to colon cancer cells with rapid internalization with no affinity toward normal colon epithelial cells have also been investigated in vitro and at the cellular level in the presented manuscript. Although the manuscript is organized, contains some new results, and has researcher interest. The methodology adopted is quite satisfactory. Thus, I recommend that the article can be published after the authors address the following points:

1. The abstract represents little confusion as line 27 described the smart enzyme responsive nanoparticles loaded with EUG and line 32 casein-coated NPs-EUG. It would be better to merge or rephrase these lines with a developed smart enzyme-responsive drug delivery system such as casein-coated EUG mesoporous silica nanoparticle (EUG/CAS-MSNs-COOH) for……    

2. Number of literature related to the delivery of EUG using nanoformulations have also already been published. So, the authors should justify how this formulation is more suitable rather than other developed formulations, and a comparison should be included in the introduction part.

3. Authors should double-check the manufacturer name of instruments used in the method section which should be incorporated in the revised manuscript.

4. Authors should mention the equation number and include it in the revised version of the manuscript.                  

5. Was the authors check the stability of eugenol in the release media having trypsin?  Please justify.

6. In Section 3.5, the Author should represent the results of particle size, zeta potential, and PDI in mean±SD for the optimized version of the formulation.

7. Authors should include appropriate references to strengthen the statement in line 461 nano assemblies gained more negatively charged groups after the introduction of carboxy groups and coupling with casein, which was reflected in their zeta-potentials.

8. In figure 6 B, it would be advantageous to include the XRD pattern of plain EUG.

9. Reviewers encourage to authors if they include in vivo Pharmacokinetics experiments in rats to strengthen the outcomes of the present study and demonstrate that the developed formulation may overcome the therapeutic limitation of EUG due to metabolism and excretion.

10. Authors should elaborate more about the future perspective of these NPs (EUG/CAS-MSNs-COOH) in the conclusion section.

11. There are minor grammatical and typo errors throughout the manuscript which should be corrected.

Author Response

The authors (Wijewantha et al.,) reported the manuscript entitled Enhancing Anti-tumorigenic Efficacy of Eugenol in Human Colon Cancer Cells Using Enzyme-Responsive Nanoparticles. Several natural plant-based products, and their active secondary metabolites, have been successfully employed as anticancer compounds as effectively target cancer cells by different signaling pathways and cellular mechanisms. While selective delivery and release of these plant-based anti-cancer compounds are still challenging. Eugenol (EUG), is one of the plant-based anti-cancer compounds which demonstrated diverse apoptotic and anti-growth properties against malignant tumors, including colorectal cancer cells. However, the rapid metabolism, excretion, and side effects on normal cells at higher dosages limit its therapeutic application in the treatment of colon cancer. In quest of that authors successfully fabricated EUG-loaded enzyme-responsive nanoparticle (eNPs) and their high affinity to colon cancer cells with rapid internalization with no affinity toward normal colon epithelial cells have also been investigated in vitro and at the cellular level in the presented manuscript. Although the manuscript is organized, contains some new results, and has researcher interest. The methodology adopted is quite satisfactory. Thus, I recommend that the article can be published after the authors address the following points:

  1. The abstract represents little confusion as line 27 described the smart enzyme responsive nanoparticles loaded with EUG and line 32 casein-coated NPs-EUG. It would be better to merge or rephrase these lines with a developed smart enzyme-responsive drug delivery system such as casein-coated EUG mesoporous silica nanoparticle (EUG/CAS-MSNs-COOH) for……    

We edited line 32 to bring the notation to the common “eNPs-EUG” term.

  1. Number of literature related to the delivery of EUG using nanoformulations have also already been published. So, the authors should justify how this formulation is more suitable rather than other developed formulations, and a comparison should be included in the introduction part.

We added references for the delivery of eugenol by lipid, tannic acid-crossed-linked, calcium citrate, and calcium carbonate-based nanoparticles. We noted that neither of those nano formulations releases eugenol when triggered with an enzyme.

  1. Authors should double-check the manufacturer name of instruments used in the method section which should be incorporated in the revised manuscript.

We refreshed/corrected the manufacturer name of instruments employed in this study in the main text.

  1. Authors should mention the equation number and include it in the revised version of the manuscript.  

We fixed this concern in the main text.               

  1. Was the authors check the stability of eugenol in the release media having trypsin?  Please justify.

Eugenol stability was tested in the released medium, and results were added to the supplementary document.

  1. In Section 3.5, the Author should represent the results of particle size, zeta potential, and PDI in mean±SD for the optimized version of the formulation.

The particle size, zeta potential and PDI values of the optimized by the content of casein nanoassembly are now presented on lines 473-478, and their optimization is emphasized on line 468.

  1. Authors should include appropriate references to strengthen the statement in line 461 nano assemblies gained more negatively charged groups after the introduction of carboxy groups and coupling with casein, which was reflected in their zeta-potentials.

The authors entirely agree with this concern. We added a reference to strengthen the statement.

  1. In figure 6 B, it would be advantageous to include the XRD pattern of plain EUG.

Figure 6B now shows XRD patterns of the materials in a periodic order to characterize their identity and morphology. Under the conditions of the experiment (room temperature and aqueous solution), we did not expect a periodic arrangement of sub-nanometer molecules of eugenol, whose presence was confirmed by UV-spectroscopy.

  1. Reviewers encourage to authors if they include in vivo Pharmacokinetics experiments in rats to strengthen the outcomes of the present study and demonstrate that the developed formulation may overcome the therapeutic limitation of EUG due to metabolism and excretion.

As reviewer #1 suggested, determining NPs-EUG’s pharmacodynamics in animal models is a critical step to verifying the safety and efficacy of NPs-EUG. An ongoing project in our group focuses on the PK/PD of NPs-EUG in two mouse models of colorectal cancer (CRC): 1) An APC CRC genetic model and 2) A spleen-liver CRC metastatic mouse model. We hope the animal experiments will be completed by the end of 2023 to be included in the next publication.  

  1. Authors should elaborate more about the future perspective of these NPs (EUG/CAS-MSNs-COOH) in the conclusion section.

We further extended the future directions and goals in the conclusion section. We will prioritize determining the underlying mechanism of EUG released by NPs in animal models. Future animal models will illustrate whether the NPs-EUG can dominantly function against metastatic tumors developed in the liver (secondary sites). Besides NPs-EUG treatment, animals with CRC (primary and metastatic models) will be treated with the combination therapy of NPs-EUG and traditional chemotherapies to determine if we can observe a synergistic effect. A synergistic effect of NPs-EUG combined with a chemotherapeutic agent can decrease the dosage of chemotherapeutic agents used in patients and lessen the incidence of drug resistance.

  1. There are minor grammatical and typo errors throughout the manuscript which should be corrected.

We double-checked the manuscript for typos, and grammatical errors, and corrected them.

Reviewer 2 Report

 Wijewantha and collaborators analysed the anti-tumorigenic activities of Eugenol in its free form or nanoencapsulated into enzyme responsive nanoparticles. The study is well designed and the experiment performed are well designed and realized. Nevertheless there are some points that have not been sufficiently clarified and that need to be more focused. In particular:

1)      Why use casein? What is the rationale?

2)      The discussion lacks a realistic analysis of how to use this type of nanoparticles in vivo. Are the authors thinking of bloodstream, oral administration or via enema? The digestive tract is full of proteases: how do they plan to reach the primary site of the CRC with their eNPs avoiding the action of proteases of human origin and of derived from the intestinal microbiota?

3)      Figure 10 has to be replaced. Panel A (WB) if the comparison is between FHC and HCT cells treated or not with Eugenol or eNP-Eugenol, why HT-29 cells are present? b-tubulin expression is very different between lane 1 and lane 2 or 3. Please load the same amount of proteins in the gel lanes. Panel C-D why absorbance and not a cell number. In Panel D I understand that Eugenol do nothing. Only casein makes differences. Same for panel F: I understand from figure that Eugenol is less toxic than nothing, and casein is very toxic. Eugenol counteracts and cancels the toxicity of casein. Is thi the message of this figure. I don’t think so.

4)      Information on the pharmacokinetics of eugenol is lacking. See for example: nt J Mol Sci. 2023 Jan 16;24(2):1800. doi: 10.3390/ijms24021800.

Author Response

Wijewantha and collaborators analysed the anti-tumorigenic activities of Eugenol in its free form or nanoencapsulated into enzyme responsive nanoparticles. The study is well designed and the experiment performed are well designed and realized. Nevertheless there are some points that have not been sufficiently clarified and that need to be more focused. In particular:

1)      Why use casein? What is the rationale?

Casein plays the role of a “gate-keeping element”, which seals eugenol inside nanoparticles and releases it when digested by the MMP-7 enzyme overproduced by cancerous cells. An additional explanation is added to lines 97-100.

2)      The discussion lacks a realistic analysis of how to use this type of nanoparticles in vivo. Are the authors thinking of bloodstream, oral administration or via enema? The digestive tract is full of proteases: how do they plan to reach the primary site of the CRC with their eNPs, avoiding the action of proteases of human origin and of derived from the intestinal microbiota?

As reviewer #2 highlighted, we reviewed all reported routes in the literature to develop and design the best administration route for the future pre-clinical model using NPs-EUG. Based on the structure and composition of casein-coated NPs and the low level of MMP-7 in plasma compared to the tumor sites (see the cited references in the main text), we have used the intravenous (IV) injection in our pilot CRC animal model. Most current chemotherapeutic agents used in CRC patients, such as 5-FU (5-fluorouracil), are typically given as an injection into the vein, as a slow IV push, or as an infusion. Additionally, postmortem experiments on the treated CRC mouse models supported the IV injection to deliver intact NPs-EUG to the tumor sites. Certainly, future modifications of NPs-EUG, such as the replacement of casein with fibronectin, a known in vivo MMP-7 substrate [1], can potentially open other potential administration routes besides IV, such as direct injection to tumor mass under colonoscopy guides. As previously described [2], we expect diverse therapeutic effects using different routes in the animal models. We added a summarized paragraph to the discussion to further describe the future animal works for this current study.

3)      Figure 10 has to be replaced. Panel A (WB) if the comparison is between FHC and HCT cells treated or not with Eugenol or eNP-Eugenol, why HT-29 cells are present? HT-29 colon cancer cells exhibit an enhanced metastatic potential, and therefore we used HT-29 cells as a positive control next to HCT-116 and FHC cells. b-tubulin expression is very different between lane 1 and lane 2 or 3. Please load the same amount of proteins in the gel lanes. We agree with the reviewer’s suggestions, and we revised Fig 10 accordingly. 

Panel C-D why absorbance and not a cell number? Unfortunately, we could not access a proper counting software to count the entire cells per plates. We used a crystal assay protocol (Abcam Crystal violet Assay Kit or Cell viability, catalog number#ab232855) to compare the entire stained cells per plate between groups. The measurement of absorption of the entire plates after staining can provide higher accurate reading than the manual counting of the cells under a microscope.

provided by In Panel D I understand that Eugenol do nothing. Only casein makes differences. Same for panel F: I understand from figure that Eugenol is less toxic than nothing, and casein is very toxic. Eugenol counteracts and cancels the toxicity of casein. Is thi the message of this figure. I don’t think so.

We are sorry for not providing a clear explanation for the observed effects in Figure 10. Blue and red columns in panel D indicate pure EUG has no toxic effect on normal cells. On the other hand, blue and red columns in panel F show pure EUG is toxic for cancer cells. The results illustrated in panel F matches with results in Figure 2 to 4. Moreover, delivery of EUG with casein-coated NPs (Green column in panel F), further decreases the cell viability in comparison to the red column (pure EUG) in cancer cells. Basically, the order of toxicity in cancer cells is as follow: NPs-EUG > pure EUG> control group> Casein-coated NPs without EUG.  

On the other hand, we surprisingly observed the casein coated NPs without EUG improved the cell viability (probably through cell adhering mechanism) in both cancer and normal cells (Purple column in both panels D and F). Expectedly, the cell viability increases in FHC cells in the presence of casein coated NPs-EUG (Green column in panel D). Since EUG alone is not toxic to normal cell and therefore combination of casein NPs and EUG did not reveal a significant toxicity impact in normal cells. We have improved the corresponding paragraphs for this section in the main text including the Figure 10’s legend.

4)      Information on the pharmacokinetics of eugenol is lacking. See for example: nt J Mol Sci. 2023 Jan 16;24(2):1800. doi: 10.3390/ijms24021800.

The requested information is now added to the manuscript (lines 70-71).

Reviewer 3 Report

In these interesting study, Authors provides data on the effect of Eugenol on CRC and on metastatic cells.  They also develop a smart delivery system to maximize the release of Eugenol at the tumor level, that could have very interesting developments. The methods used are appropriate.

Minor

In section 3.7 Authors states that “Casein-coated NPs-EUG decreases cell viability in cancer cells while normal colon epithelial 576 cells remain intact”.  

HCT-116 colon cancer and FHC normal colon epithelial cells were both treated with

1)      Free eugenol (pure EUG)

2)      EUG delivered by casein-coated NPs

3)      casein-coated NPs

Authors performed a crystal violet cell cytotoxicity assay, but the way results are presented in the figure is confusing, and should be improved.

Authors says that pure EUG had no  effect on FHC cells but significantly decreased the viability of HCT-116 cells ( in panel D and F values are reported once in absorbance ( D) and once in cytotoxicity fold change (F), and this should be uniformed)

In the same panels, values for treatment with  EUG delivered by casein-coated NPs and casein-coated NPs are also reported. Authors says that casein increased the number of adherent cells, but in this case a value more similar to control group was expected in panel D. Is it possible that casein interfered with assay?

In the Panel F, scale reports “Cytotoxicity fold change”. Does values ranging from 0.0 to 2.0 refers to an increase in cytotoxicity ? Basing on how the data are depicted, EUG delivered by casein-coated NPs appear to be less cytotoxic than the other tratments, including control group. Moreover, a fold change of 0 in toxicity should have been assigned to the control group.

Author Response

In these interesting study, Authors provides data on the effect of Eugenol on CRC and on metastatic cells.  They also develop a smart delivery system to maximize the release of Eugenol at the tumor level, that could have very interesting developments. The methods used are appropriate.

Minor

In section 3.7 Authors states that “Casein-coated NPs-EUG decreases cell viability in cancer cells while normal colon epithelial 576 cells remain intact”. 

HCT-116 colon cancer and FHC normal colon epithelial cells were both treated with

1)      Free eugenol (pure EUG)

2)      EUG delivered by casein-coated NPs

3)      casein-coated NPs

Authors performed a crystal violet cell cytotoxicity assay, but the way results are presented in the figure is confusing, and should be improved.

Authors says that pure EUG had no  effect on FHC cells but significantly decreased the viability of HCT-116 cells ( in panel D and F values are reported once in absorbance ( D) and once in cytotoxicity fold change (F), and this should be uniformed).

We thank reviewer for highlighting this mistake. Both panel D and F show absorbance and we fixed this issue in Figure 10.

In the same panels, values for treatment with  EUG delivered by casein-coated NPs and casein-coated NPs are also reported. Authors says that casein increased the number of adherent cells, but in this case a value more similar to control group was expected in panel D. Is it possible that casein interfered with assay?

We agree with Reviewer # 3 regarding casein and its possible interfering effect on this current crystal assay. The casein-coated NPs-EUG was able to successfully deliver its toxicity to cancer cells while it failed to similarly perform in normal cells. Therefore, EUG is the dominant factor when we used casein coated NPs-EUG on cancer cells. We believe the improvement of cell viability in normal cells and cancer cells (purple column) by casein is heavily mediated by the effect of casein on cell adhesion while we cannot eliminate the casein’s crosstalks with other cell component or reagents in assay kit. We explain/clarify this concern in the main text.

In the Panel F, scale reports “Cytotoxicity fold change”. Does values ranging from 0.0 to 2.0 refers to an increase in cytotoxicity ? Basing on how the data are depicted, EUG delivered by casein-coated NPs appear to be less cytotoxic than the other tratments, including control group. Moreover, a fold change of 0 in toxicity should have been assigned to the control group.

The confusion is due to misslabeling of Y axis in panel D and F which are both absorbance. We fixed this issue in the revised Figure 10. As reviewer #3 highlighted, the level of absorption was generally higher in FHC normal cell (Panel D) in comparison to cancer cells (Panel F). This was expected since FHC cells are much larger than HCT-116 cells and they provide more cell surface for staining. We used the same number of cells for both cell lines. The corresponding paragraphs in the main text have been revised accordingly.  

Reviewer 4 Report

Overall the study is carried out nicely, with a logical conclusion. However, various latest references (2020-2023) were missing on the topic, which I think should be included before formally accepting this paper for publication. 

I also wish to see the relevant reference of other molecules, engineered as casein-coated mesoporous silica nanoparticles that efficiently encapsulate other compounds besides EUG in the discussion section. 

Moreover, the conclusion needs to be tried to have a specific conclusion. 

Author Response

Overall the study is carried out nicely, with a logical conclusion. However, various latest references (2020-2023) were missing on the topic, which I think should be included before formally accepting this paper for publication.

The latest references on the delivery on pharmacokinetics of eugenol and its delivery by lipid, tannic acid-crossed-linked, calcium citrate, and calcium carbonate-based nanoparticles have been added to the manuscript.

I also wish to see the relevant reference of other molecules, engineered as casein-coated mesoporous silica nanoparticles that efficiently encapsulate other compounds besides EUG in the discussion section.

The application of casein-coated mesoporous silica nanoparticles to the targeted delivery of veratridine is now referenced in the manuscript (ref. 46).

Moreover, the conclusion needs to be tried to have a specific conclusion.

We further improved content of the conclusion section and we included an extended future work direction.

Round 2

Reviewer 2 Report

I acknowledge that the requested changes have been made.